# Structural basis of HEAT-kleisin interactions in the human condensin I subcomplex

Kodai Hara[1,*] ID, Kazuhisa Kinoshita[2], Tomoko Migita[1], Kei Murakami[1], Kenichiro Shimizu[1], Kozo Takeuchi[2,†], Tatsuya Hirano[2,**] ID & Hiroshi Hashimoto[1] ID

## Abstract

Condensin I is a multi-protein complex that plays an essential role in mitotic chromosome assembly and segregation in eukaryotes. It is composed of five subunits: two SMC (SMC2 and SMC4), a kleisin (CAP-H), and two HEAT-repeat (CAP-D2 and CAP-G) subunits. Although balancing acts of the two HEAT-repeat subunits have been demonstrated to enable this complex to support the dynamic assembly of chromosomal axes in vertebrate cells, its underlying mechanisms remain poorly understood. Here, we report the crystal structure of a human condensin I subcomplex comprising hCAP-G and hCAP-H. hCAP-H binds to the concave surfaces of a harp-shaped HEAT-repeat domain of hCAP-G. Physical interaction between hCAP-G and hCAP-H is indeed essential for mitotic chromosome assembly recapitulated in *Xenopus* egg cell-free extracts. Furthermore, this study reveals that the human CAP-G-H subcomplex has the ability to interact with not only double-stranded DNA, but also single-stranded DNA, suggesting functional divergence of the vertebrate condensin I complex in proper mitotic chromosome assembly.

**Keywords** chromosome condensation; HEAT repeats; HEAT-kleisin interaction; ssDNA binding; X-ray crystallography
**Subject Categories** Cell Cycle; Structural Biology

## Introduction

Immediately before cell division, chromatin that resides in the nucleus is converted into a set of rod-shaped structures to support their faithful segregation into daughter cells. The condensin complexes play a central role in this process, known as mitotic chromosome assembly or condensation, and also participate in diverse chromosome functions such as gene regulation, recombination, and repair [1,2]. Moreover, hypomorphic mutations in the genes encoding condensin subunits have been implicated in the human disease

microcephaly [3]. Many eukaryotes have two different types of condensin complexes, namely, condensins I and II. Condensin I, for example, consists of a pair of structural maintenance of chromosomes (SMC) ATPase subunits (SMC2 and SMC4) and three non-SMC regulatory subunits (CAP-D2, CAP-G, and CAP-H). SMC2 and SMC4 dimerize through their hinge domains to form a V-shaped heterodimer, and CAP-H, which belongs to the kleisin family of proteins, bridges SMC head domains through its C- and N-terminal regions. CAP-D2 and CAP-G, both of which are composed of arrays of short amphiphilic helices known as HEAT repeats, bind to the central region of CAP-H [4,5]. Although many if not all prokaryotic species have a primitive type of condensin composed of an SMC homodimer and two other regulatory subunits including a kleisin subunit, the HEAT-repeat subunits are unique to eukaryotic condensins and not found in prokaryotic condensins.

Biochemical studies using purified condensin I holocomplexes identified several ATP-dependent activities *in vitro*, such as positive supercoiling of DNA [6–8], DNA compaction [9], translocation along dsDNA [10], and DNA loop extrusion [11]. Mechanistically, how these activities are supported by condensin I remains poorly understood. Indeed, condensin I can interact with DNA in many different ways. For instance, like cohesin and prokaryotic SMC complexes, it encircles double-stranded DNA (dsDNA) within its tripartite ring composed of the SMC dimer and kleisin [12–14]. It has also been reported that a mouse SMC2-SMC4 hinge domain binds single-stranded DNA (ssDNA), but not dsDNA [15], whereas a budding yeast non-SMC subcomplex composed of YCG1/CAP-G, YCS4/CAP-D2, and BRN1/CAP-H binds dsDNA, but not ssDNA [16]. A recent study reported the crystal structure of a budding yeast non-SMC subcomplex consisting of YCG1 and BRN1 bound to dsDNA [17]. Another study using *Xenopus* egg cell-free extracts found that the pair of HEAT-repeat subunits plays an essential role in the dynamic assembly of mitotic chromosome axes [18].

In the current study, we determined the crystal structure of a human subcomplex composed of CAP-G bound by a short fragment of CAP-H. The structure established molecular interactions between human CAP-G and CAP-H, and implicated these interactions in the ability of condensin I to support mitotic chromosome assembly. Furthermore, the human CAP-G-H subcomplex bound both dsDNA

1   Department of Physical Biochemistry, School of Pharmaceutical Sciences, University of Shizuoka, Shizuoka, Japan
2   Chromosome Dynamics Laboratory, RIKEN, Wako, Saitama, Japan
    *Corresponding author. Tel: +81 54 264 5646; Fax: +81 54 264 5644; E-mail: khara@u-shizuoka-ken.ac.jp
    **Corresponding author. Tel: +81 48 467 9531; Fax: +81 48 462 4673; E-mail: hiranot@riken.jp
    †Present address: Hamamatsu Photonics K. K., Hamamatsu, Shizuoka, Japan

and ssDNA, suggesting the functional divergence of the eukaryotic condensin I complex.

## Results and Discussion

### Structure of the human CAP-G-H subcomplex

The consensus sequence of HEAT repeats at the primary structure level is not tight. The original report by Neuwald and Hirano [19] assigned nine HEAT repeats in vertebrate CAP-G, whereas a subsequent re-assignment by Yoshimura and Hirano [5] identified 19 HEAT repeats that span the near-entire length of human CAP-G (hCAP-G). Furthermore, the secondary structural prediction server PrDOS [20] predicted that hCAP-G has two long disordered regions (amino acid residues 477–553 and 896–1,015) and five short disordered regions (residues 1–12, 81–93, 382–393, 660–687, and 812–821) (Fig 1A, upper). On the other hand, human CAP-H (hCAP-H) has five regions that are conserved among its orthologs among eukaryotic species (motifs I-V) (Fig 1A, lower). A previous biochemical study revealed that the N-terminal and C-terminal halves of hCAP-H bind to hCAP-D2 and hCAP-G, respectively [4]. As the most C-terminally located motif V was predicted to bind to SMC2 [21], we thought that motif IV (residues 461–503) may be responsible for binding to hCAP-G. With this information, we aimed to express and purify hCAP-G complexed with a fragment of hCAP-H. We found that the N-terminal domain of hCAP-G (residues 1–478) connected to the C-terminal domain of hCAP-G (residues 554–900), and a fragment of hCAP-H containing motif IV (residues 460–515) was able to be co-expressed and co-purified (Fig 1B). This hCAP-G-H subcomplex was successfully crystalized and its structure was determined at 3.0 Å resolution (Table 1). Two molecules of the hCAP-G-H subcomplex are present in the crystallographic asymmetric unit (Fig EV2A). Their structures are essentially identical, but 4-(2-hydroxyethyl)-1-piperazineethanesulfonic acid (HEPES) is bound to only one of the two molecules. In the current report, we describe the HEPES-bound hCAP-G-H subcomplex (a, b-molecules) as a representative structure (Fig 1C). Consistent with the recent assignment based on its amino acid sequence [5], hCAP-G displays a "harp-shaped" structure composed of 19 HEAT repeats (H1-H19), in which H12 and H15 have long disordered loops (residues 479–553 and 661–691, respectively) (Figs 1C and EV1A and EV2B). hCAP-H, which comprises three α-helices (α2, α3′, and α4), binds to the concave surfaces of hCAP-G (Figs 1C and EV1B). This overall structure in which a kleisin fragment binds to the concave surfaces of a harp-shaped HEAT-repeat domain is highly reminiscent of other cohesin subunits and its regulators [22–24], as well as budding and fission yeast condensin subunits (YCG1-BRN1 and CND3/CAP-G-CND2/CAP-H) [17]. It should be noted that the hCAP-G used in this study shares only 16 and 21% amino acid identity with YCG1 and CND3, respectively, and that the hCAP-H fragment bound to hCAP-G shares only 25 and 29% identity with BRN1 and CND2, respectively. Although there is great divergence in their amino acid sequences, two basic residues (K60 and R848) located at the N- and C-terminal lobes of hCAP-G, which correspond to DNA-binding residues K70 (YC1) and R849 (YC2) of YCG1, respectively, are structurally well conserved (Fig EV1A). Similarly, four basic residues (R435, R437, K456, and K457) of hCAP-H, which correspond to K409 (BC1), R411 (BC1), K456 (BC2), and K457 (BC2) of BRN1, respectively, are also conserved, but we were unable to visualize these residues because they were not included in the crystallized recombinant protein (Fig EV1B). Kschonsak *et al* [17] recently demonstrated that the corresponding amino acid residues of YCG1-BRN1 function in dsDNA binding, and proposed a "safety-belt mode" by which a peptide loop produced by two regions of BRN1, namely a latch and buckle, encircles the bound DNA and prevents its dissociation. The previous study strongly suggests that the hCAP-G-H subcomplex also uses these residues to bind to dsDNA (see below).

We next performed superimpositions between the hCAP-G-H subcomplex and its budding/fission yeast counterparts, YCG1-BRN1/CND3-CND2, using PyMoL (http://www.pymol.org/). Structural alignments between hCAP-G-H and YCG1-BRN1/CND3-CND2 each had a root mean square deviation (RMSD) value of 4.293 and 5.272 Å for 3,990 and 4,049 superimposable atoms, respectively (Fig 1D, orange and blue or green), whereas the RMSD value between YCG1-BRN1 and CND3-CND2 was 3.145 Å for 8,075 superimposable atoms (Fig 1D, blue and green). These superimpositions revealed that the main chain structure of hCAP-G-H is different from that of its yeast counterpart, explaining why we were unable to determine the structure of hCAP-G-H by molecular replacement using its yeast counterpart structures. In addition, we performed superimpositions between DNA-bound YCG1-BRN1 and DNA-free forms. The RMSD value between the DNA-bound YCG1-BRN1 and hCAP-G-H was 5.316 Å for 4,126 superimposable atoms (Fig 1E, red and orange), whereas the RMSD value between DNA-bound YCG1-BRN1 and YCG1-BRN1/CND3-CND2 was 2.062 and 2.732 Å for 12,997 and 8,199 superimposable atoms, respectively (Fig 1E, red

**Figure 1.  Domain architecture and overall structure of the hCAP-G-H subcomplex.**

A    hCAP-G is composed of 1,015 amino acids and contains 19 HEAT repeats. hCAP-H is composed of 730 amino acids and contains 5 conserved motifs (I: hSMC2 binding region, II: hCAP-D2 binding region, III: DNA-binding region, IV: hCAP-G binding region, V: hSMC4 binding region).

B    Scheme of the hCAP-G-H subcomplex. Residues 1–478 of hCAP-G were fused to residues 554–900 of hCAP-G. hCAP-G (1–478, 554–900) was co-expressed in *E. coli* and co-purified for crystallography.

C    Cartoon diagram of the crystal structure of hCAP-G (orange) in complex with a fragment of hCAP-H (green). Unstructured, disordered regions are indicated by the dots. The 19 HEAT repeats (H1-H19) and 2 disordered loops (H12 loop and H15 loop) of hCAP-G, and 4 helices (α2, α3′, and α4) of hCAP-H are labeled. The N- and C-termini of CAP-G and CAP-H are also indicated. A molecule of 4-(2-hydroxyethyl)-1-piperazineethanesulfonic acid (HEPES) is shown by the orange-colored stick model. A 90-degree rotated version is shown on the right.

D    Comparison of the hCAP-G-H subcomplex with its related structures. Superimposition of the structures of hCAP-G-H (orange), *S. cerevisiae* YCG1-BRN1 (PDB ID: 5OQQ; blue), and *S. pombe* CND3-CND2 (PDB ID: 5OQR; green) is presented as a Cα-tracing model.

E    Comparison of the DNA-bound form with DNA-free forms. Superimposition of the structures of DNA-bound YCG1-BRN1 (PDB ID: 5OQN; red), hCAP-G-H (orange), YCG1-BRN1 (blue), and CND3-CND2 (green) is presented as in (D).

and blue or green). These superimpositions suggested that the overall structure of DNA-bound YCG1-BRN1 is identical to the structure of DNA-free YCG1-BRN1.

There are several notable differences between the human and yeast structures on comparison of our structure with the previous one. First, some secondary structures of the hCAP-G-H subcomplex

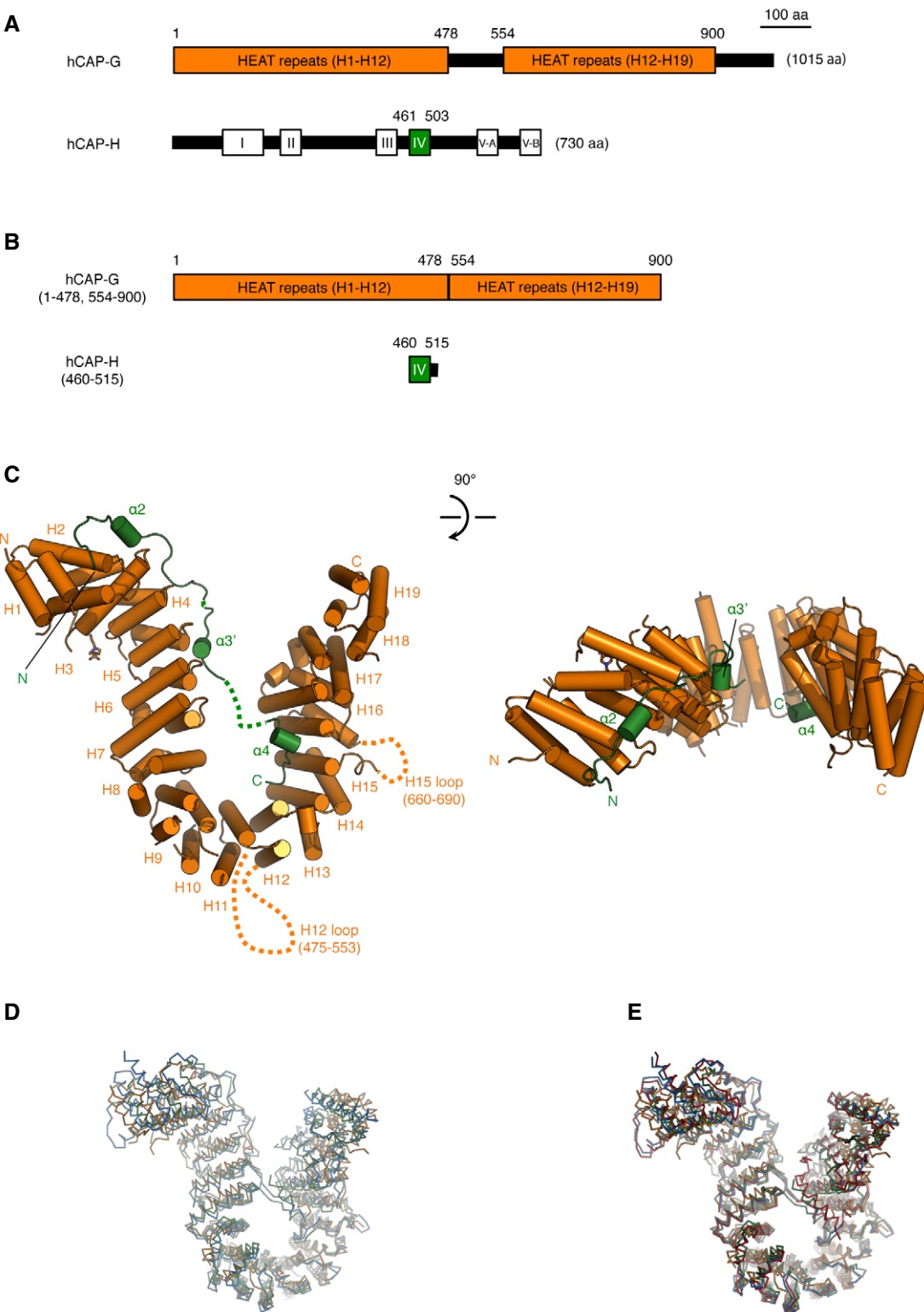

**Figure 1.**

**Table 1.    X-ray crystallography: data collection and refinement statistics.**

|  | Native | Au (peak) |
|---|---|---|
| Data collection | | |
| Space group | $P2_1$ | $P2_1$ |
| Cell dimensions | | |
| *a, b, c* (Å) | 122.4, 61.9, 130.9 | 122.5, 61.2, 131.3 |
| *α, β, γ* (°) | 90.0, 93.4, 90.0 | 90.0, 93.6, 90.0 |
| Resolution (Å) | 19.81–3.00 (3.12–3.00) | 19.73–3.38 (3.58–3.38) |
| No. total/unique reflections | 132,620/39,503 | 364,483/27,655 |
| $R_{merge}$ | 0.070 (0.600) | 0.161 (0.879) |
| $R_{pim}$ | 0.045 (0.375) | 0.046 (0.254) |
| $CC_{1/2}$ | 0.998 (0.675) | 0.998 (0.806) |
| $I/\sigma I$ | 12.7 (2.0) | 12.9 (3.3) |
| Completeness (%) | 98.8 (97.1) | 99.4 (99.1) |
| Redundancy | 3.4 | 13.2 |
| Refinement | | |
| $R_{work}/R_{free}$ | 21.2/27.1 | |
| No. atoms | | |
| Protein | 12,341 | |
| Ligand | 15 | |
| Water | 6 | |
| *B*-factors | | |
| Protein | 74.80 | |
| Ligand | 98.40 | |
| Water | 51.10 | |
| R.m.s. deviations | | |
| Bond lengths (Å) | 0.002 | |
| Bond angles (°) | 0.684 | |

Values in parentheses are for the highest resolution shell.

are different from those of the yeast counterpart. The H12 loop is a common disordered loop also found in the yeast counterpart, but the H15 disordered loop present in hCAP-G is missing in its yeast counterpart (Figs 1C and EV1A and EV2B). The hCAP-H sequence (residues 499–503), which corresponds to the buckle region of BRN1 (residues 498–504) and CND2 (residues 519–523), is also disordered in our hCAP-G-H structure (Fig EV1B). Notably, the corresponding α3 helix of BRN1 does not exist in hCAP-H. Instead, hCAP-H has the α3′ helices, producing a disordered loop that connects between the α2 and α3′ helices (Fig EV1B). Overall, the hCAP-G-H subcomplex is structurally more flexible than the YCG1-BRN1 subcomplex. Indeed, the b-factors of hCAP-G-H were higher than those of the yeast counterpart, especially the middle HEAT-repeat domain connecting the N- and C-terminal regions of hCAP-G (Fig EV2C). Second, hCAP-H is more loosely bound to hCAP-G than the yeast counterpart, resulting in the more opened conformation of the HEAT-repeat subunit hCAP-G. The distance between R257 and V754 of hCAP-G is 17.06 Å (Fig 2A), whereas the corresponding distance between R287 and F749 of YCG1 is 6.25 Å (Fig EV3A). The distance between K154 and K889 of hCAP-G is 35.29 Å (Fig 2A),

whereas the corresponding distance between R170 and K895 of YCG1 is 23.02 Å (Fig EV3A). The previous electron microscopy study also demonstrated that the HEAT-repeat subunit of cohesin loader, Scc2, adopted several flexible conformations [25]. These observations of the structural flexibility led us to speculate that our hCAP-G-H structure represents a snapshot of an "open conformation", whereas the structure of its yeast counterpart represents a snapshot of a "closed conformation".

## Structural details of the interaction between hCAP-G and hCAP-H

hCAP-H interacts extensively with the concave surface of hCAP-G at four major sites (Fig 2A), whereas YCG1 interacts with BRN1 at five major sites (Fig EV3A) [17]. At site I, a pocket of hCAP-G accommodates I461 and F463 of hCAP-H in a hydrophobic manner (Fig 2B). F469, Y472, and F473 of hCAP-H make mainly hydrophobic interactions with hCAP-G. In the YCG1-BRN1 subcomplex, I461, F463, E471, V474, and F475 of BRN1 form conserved hydrophobic interactions with YCG1 (Fig EV3B).

At site II, hCAP-H interacts with hCAP-G by van der Waals forces. K475, T476, A479, T480, and I481 of CAP-H are accommodated in a shallow pocket of CAP-G (Fig 2C). At site II of the YCG1-BRN1 subcomplex, some residues (K478, T481, K482, I483, D484, and M485) of BRN1 also interact with YCG1 by van der Waals interactions. In particular, I483 and M485 of BRN1 are accommodated in the corresponding pockets of YCG1 (Fig EV3C). The site II interactions in both hCAP-G-H and YCG1-BRN1 primarily involve hydrophobic interactions, but the depths of their interaction pockets are substantially different: hCAP-G recognizes the small side chain of hCAP-H, whereas YCG1 may recognize the bulky side chains of BRN1.

At site III, T495 of hCAP-H is accommodated in a shallow pocket of hCAP-G (Fig 2D). A pocket of hCAP-G accommodates L497 of hCAP-H through van der Waals contacts. Notably, W492 of hCAP-H interacts with R493 on the same helix. This interaction may stabilize the binding of W492 of hCAP-H to E188 of hCAP-G mediated by van der Waals forces. At site III of the YCG1-BRN1 subcomplex, YCG1 interacts with BRN1 by van der Waals interactions. R490 of BRN1 interacts with Y168 of YCG1 and Y496 of BRN1 forms a hydrophobic interaction with P218 of YCG1 (Fig EV3D). K491, H495, and L497 of BRN1 are accommodated in an elongated cleft of YCG1. Site III of the YCG1-BRN1 subcomplex includes deeper clefts than that of hCAP-G, enabling YCG1 to bind bulky residues of BRN1. Differences in site III may explain why amino acid sequences between hCAP-H and BRN1 are not well conserved.

At site IV, N504, V505, L508, and V509 of hCAP-H are accommodated in a pocket of hCAP-G (Fig 2E). Five residues (L511, H512, L513, K514, and P515) of hCAP-H are also accommodated in an elongated cleft of hCAP-G. I509, F513, and I514 of BRN1 corresponding to L508, H512, and L513 of hCAP-H also interact with YCG1 in a hydrophobic manner (Fig EV3F). At site IV, there are notable hydrogen bonds formed between H512 and L513 of hCAP-H and D647 of hCAP-G (Fig 3A). D647 is an acidic residue broadly conserved among the CAP-G/YCG1 orthologs (Fig EV1A). Of note, an aspartate side chain that makes hydrogen bonds with two backbone amides of residues in a pocket is commonly found in the prefusion state of hemagglutinin (HA) of the influenza virus [26], and a binding hotspot between the HEAT-repeat subunit SA2 of cohesin and kleisin subunit Scc1 [22].

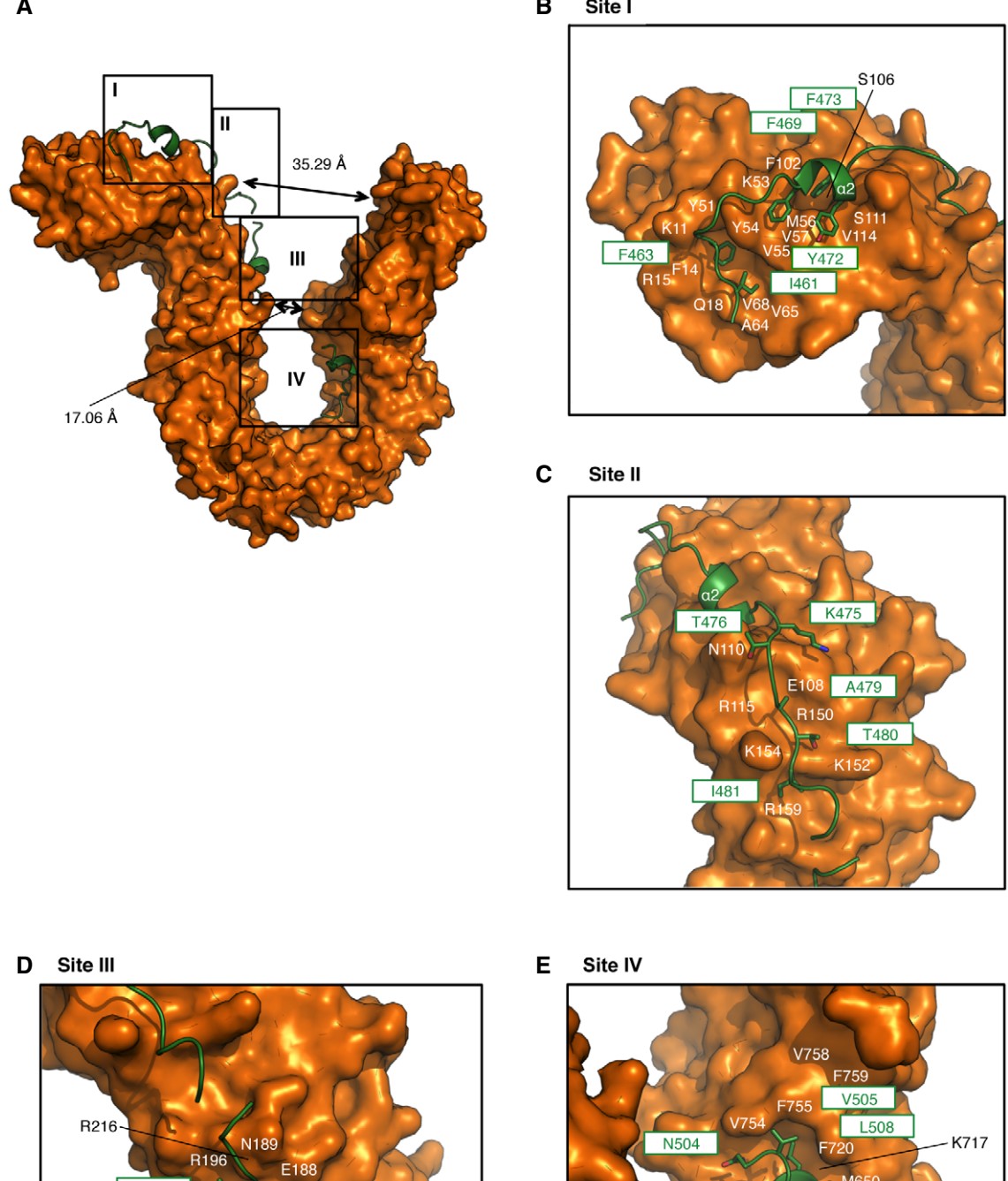

**Figure 2.  Structural details of the interaction between hCAP-G and hCAP-H.**

A    The molecular surface of hCAP-G is shown in orange. hCAP-H is shown as a green ribbon model. The four major contact sites (I, II, III, and IV) are boxed.

B–E   Zoomed-in views of sites I–IV. Residues of hCAP-G and hCAP-H are labeled in white or black and green, respectively.

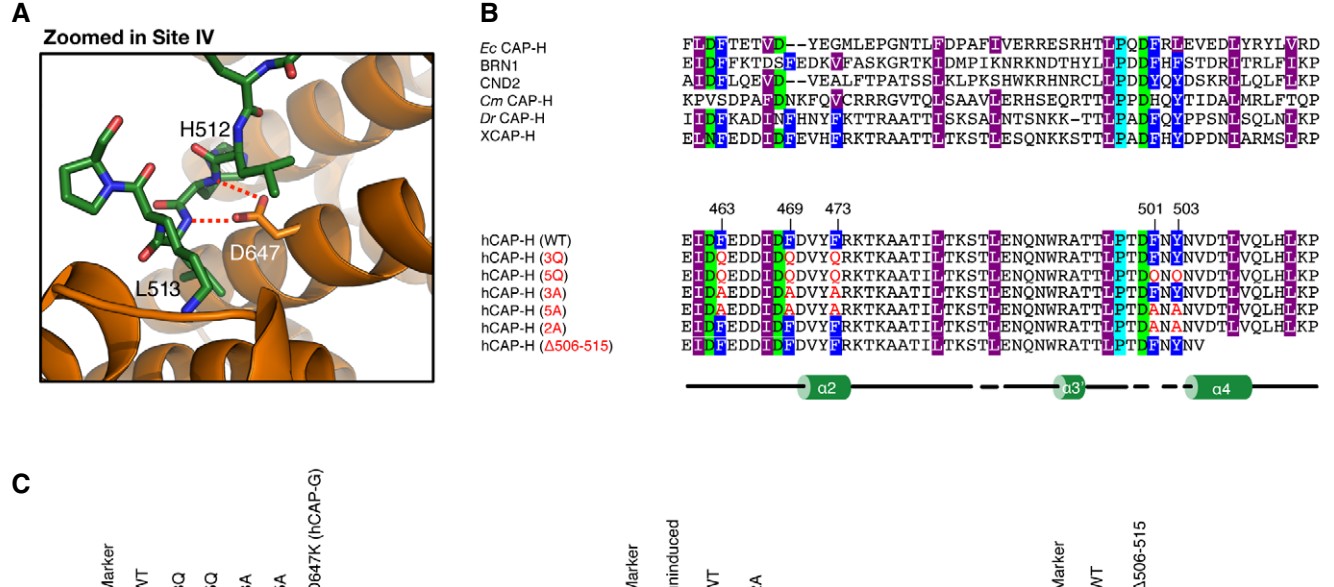

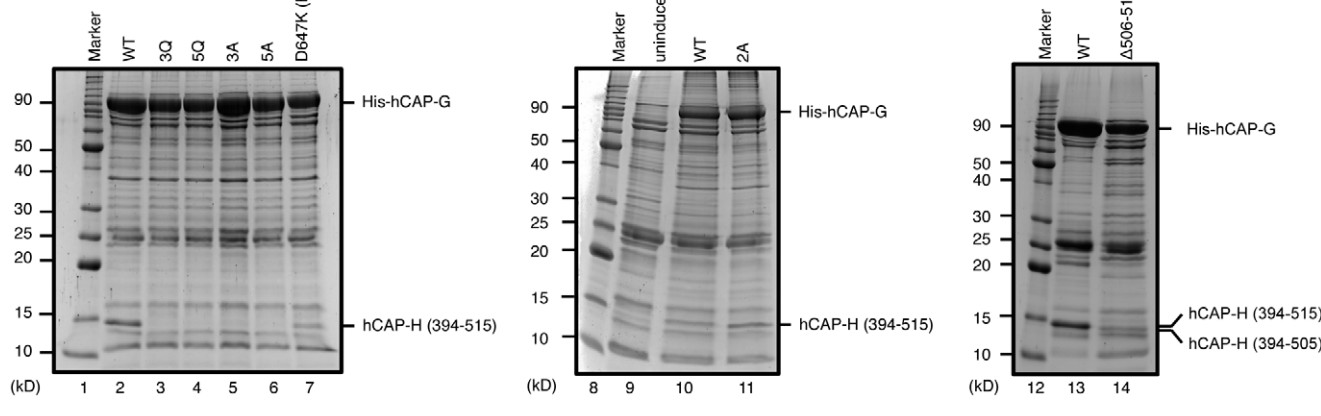

**Figure 3. Identification of residues required for interaction between hCAP-G and hCAP-H.**

A    Zoomed-in view of site IV. Residues of hCAP-G and hCAP-H are shown in orange and green, respectively. The dashed red lines indicate hydrogen bonds.

B    3Q, 5Q, 3A, 5A, 2A, and Δ506–515 mutants of hCAP-H. Motif IV (residues 461–503) contains amino acid residues highly conserved among eukaryotic species (X, *Xenopus laevis*; Dr, *Danio rerio*; Cm, *Cyanidioschyzon merolae*; Sp, *Schyzosaccharomyces pombe*; Sc, *Saccharomyces cerevisiae*; Ec, *Encephalitozoon cuniculi*). To produce the IV-3Q, 5Q, 3A, 5A, and 2A mutants, the conserved aromatic amino acid residues (F463, F469, F473, F501, and Y503; labeled in dark blue) were substituted with glutamine (Q) or alanine (A) residues. The secondary structural element of hCAP-H is drawn below the sequence alignments.

C    Interaction analysis between hCAP-G and hCAP-H. Bacterial cell lysates co-expressing hCAP-G (residues 1–478, 554–900) and hCAP-H (residues 394–515), either wild type (WT; lanes 2, 10 and 13), 3Q (F463Q, F469Q and F473Q; lane 3), 5Q (F463Q, F469Q, F473Q, F501Q and Y503Q; lane 4), 3A (F463A, F469A and F473A; lane 5), 5A (F463A, F469A, F473A, F501A and Y503A; lane 6), or 2A (F501A and Y503A; lane 11), or a C-terminal deletion mutant (506–514 residues were deleted from 394–515; lane 14) were applied to Ni-NTA agarose resin, and the bound fraction was analyzed by SDS–PAGE. Alternatively, a cell lysate co-expressing mutant hCAP-G (D647K) and wild-type hCAP-H was examined (lane 7). The uninduced cell lysate was also used as a negative control (lane 9).

The YCG1-BRN1 subcomplex has an additional HEAT-kleisin interaction site, site III′ (Fig EV3E). At site III′, L498, P499, D501, F502, and F504 of BRN1 interact with YCG1 (Fig EV3E). Although no interactions corresponding to site III′ are found in the hCAP-G-H subcomplex, F501 and Y503 of hCAP-H corresponding to F502 and F504 of BRN1 are highly conserved among eukaryotic species. It is therefore possible that the hCAP-G-H subcomplex undergoes conformational changes (from an open form to a closed form), forming the site III′ interactions found in the YCG1-BRN1 subcomplex.

## Identification of residues required for interaction between hCAP-G and hCAP-H

To identify residues required for interaction between hCAP-G and hCAP-H, we designed six mutants that targeted conserved, surface-exposed residues at the hCAP-G-H interface, and the amount of the hCAP-H fragment that co-purified with immobilized His$_6$-tagged hCAP-G was evaluated (Fig 3B and C). As expected, three Gln substitutions (3Q) of F463, F469, and F473 of hCAP-H positioned at site I greatly impaired the interaction between hCAP-G and hCAP-H (Fig 3C, lanes 2 and 3). Three Ala substitutions (3A) of the same residues also diminished the interaction, suggesting that van der Waals interactions formed by these aromatic residues of CAP-H are essential for its interaction with hCAP-G (Fig 3C, lane 5). Moreover, Ala substitutions (2A) and additional Gln or Ala substitutions (5Q or 5A) of F501 and Y503 of hCAP-H positioned at site III′ did not further impair its interaction with hCAP-G (Fig 3C, lanes 4, 6, and 11). This suggests that F501 and Y503 are not directly involved in hCAP-G-H subcomplex formation, but may be required for the stabilization of a closed conformation after dsDNA binding. A Lys

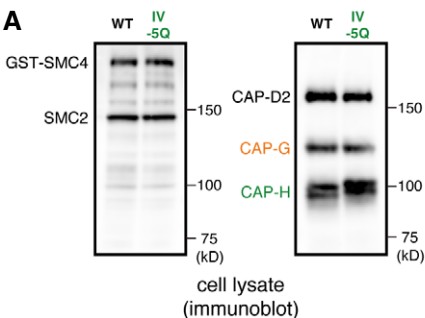

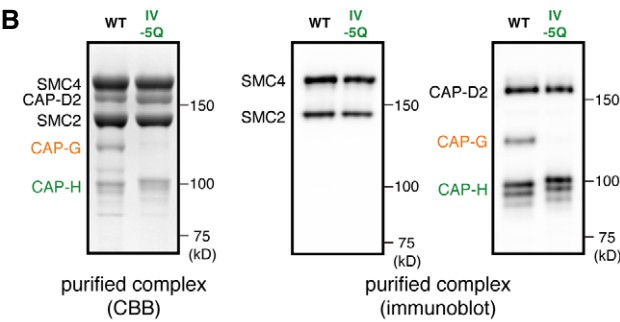

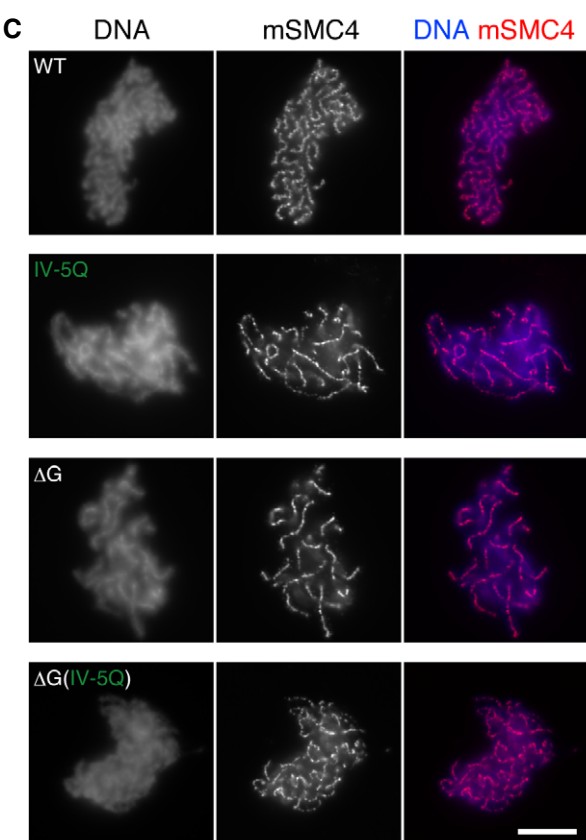

**Figure 4.  CAP-H motif IV is required for a physical interaction with CAP-G.**

A   Expression of condensin I subunits in insect cells. The wild-type (WT) or IV-5Q mutant CAP-H subunit was co-expressed with the other four subunits (GST-SMC4, SMC2, CAP-D2, and CAP-G) in insect cells. Cell lysates were prepared and subjected to SDS–PAGE, followed by immunoblotting with a mixture of antibodies against SMC2 and SMC4 (left panel) or against CAP-D2, CAP-G, and CAP-H (right panel).

B   Purification of the WT and IV-5Q mutant condensin I complexes. Protein samples purified through glutathione-affinity chromatography were subjected to SDS–PAGE and analyzed by CBB staining (left panel) or immunoblotting with a mixture of antibodies as described above (middle and right panels).

C   Add-back assay using the WT and mutant condensin I complexes. *Xenopus* extracts depleted of endogenous condensin complexes were supplemented with the purified complexes (from top to bottom; WT, IV-5Q, ΔG, ΔG[IV-5Q]). The supplemented extracts were then incubated with sperm nuclei to assemble mitotic chromosomes. The samples were fixed and labeled with an antibody against mSMC4 (red). DNA was counterstained with DAPI (blue). The data from a single representative experiment out of two repeats are shown. In the experiment shown here, multiple images were collected for condensin-depleted extracts supplemented with the WT (*n* = 17), IV-5Q (*n* = 22), ΔG (*n* = 20), and ΔG(IV-5Q) (*n* = 25) complexes. The scale bar represents 10 μm.

substitution of D647 of hCAP-G positioned at site IV (D647K) decreased its interaction with hCAP-H, suggesting that D647-mediated hydrogen bonds with H512 and L513 of hCAP-H are necessary for its interaction with hCAP-G at site IV (Fig 3A and C, lane 7). We also found that deletion of a C-terminal region of hCAP-H (506–514 residues) also reduced its interaction with hCAP-G (Fig 3C, lanes 13 and 14), supporting the idea that hydrophobic interactions formed by site IV are essential for hCAP-G-H subcomplex formation, just like site I.

**The interaction between hCAP-G and hCAP-H is essential for proper chromosome assembly mediated by condensin I in *Xenopus* egg extracts**

To test whether the interaction between hCAP-G and hCAP-H is indeed essential for the function of condensin I, we introduced the motif IV quintuple mutations (F463Q, F469Q, F473Q, F501Q, Y503Q; designated IV-5Q) described above into the context of full-length, holocomplexes (Fig 3B). Using the baculovirus expression system described previously [18], we co-expressed the five subunits of mammalian condensin I containing either the wild-type or mutant form of hCAP-H in insect cells. An equal level of expression of the five subunits in the two samples was confirmed by immunoblotting against total lysates (Fig 4A). Both lysates were then subjected to affinity purification using glutathione-agarose beads (Note that the SMC4 subunit was GST-tagged), followed by proteolytic cleavage of the GST moiety. Although wild-type hCAP-G was successfully co-purified along with the other four subunits, the IV-5Q mutant form of hCAP-G was almost completely missing from the purified fraction (Fig 4B). The complexes purified from the wild-type and mutant lysates were then added back into *Xenopus* egg extracts depleted of endogenous condensins [18]. We found that although the holocomplex purified from the wild-type lysate produced normal chromosomes (Fig 4C, WT), the complex purified from the mutant lysate failed to do so, creating abnormal chromosomes with fuzzy surfaces and thin axes (Fig 4C, IV-5Q). The abnormal structure was highly reminiscent of those produced by the tetrameric mutant complex that lacks the hCAP-G subunit, which we reported previously

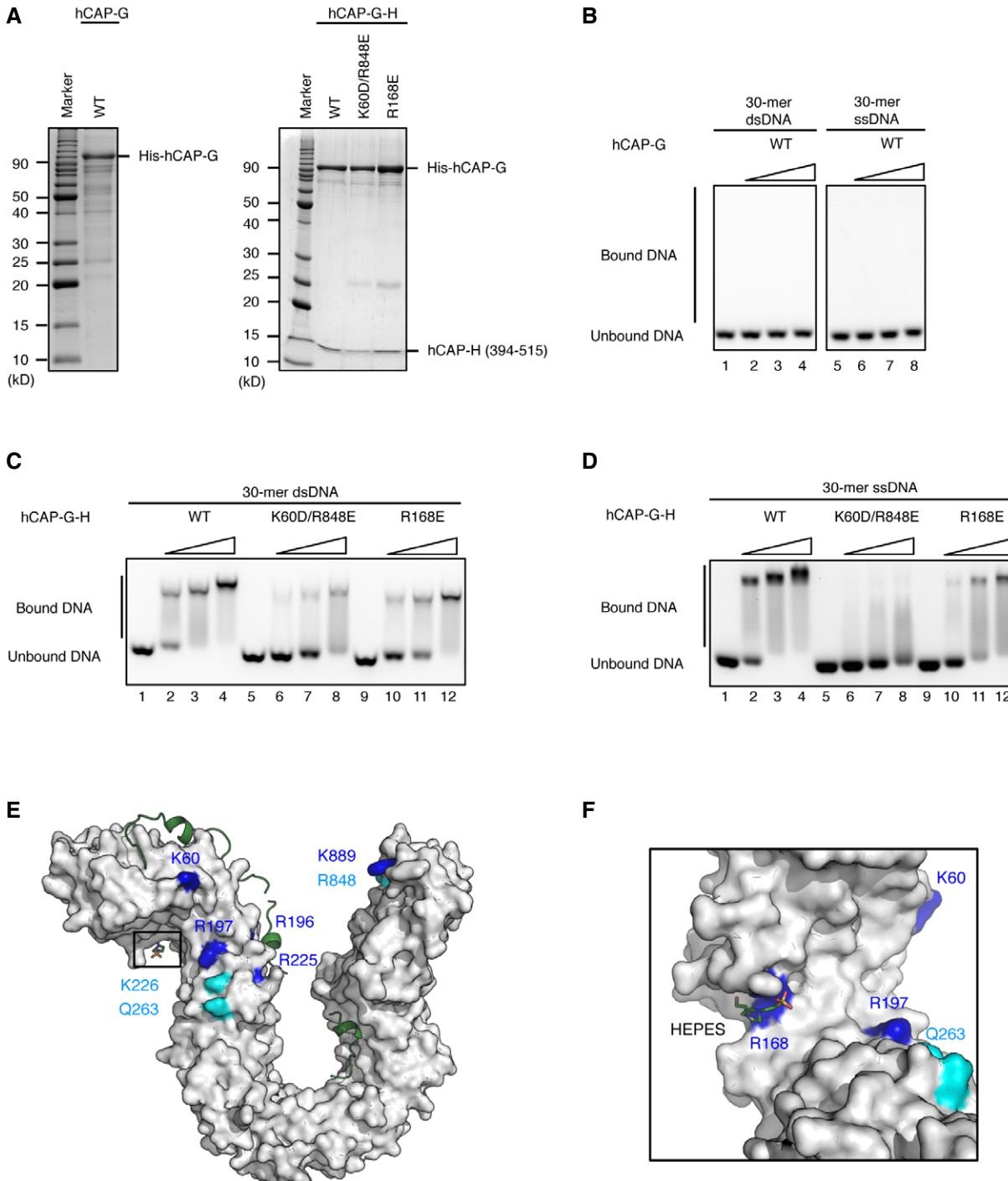

**Figure 5.   DNA-binding surfaces conserved between hCAP-G and YCG1.**

A   Purification of hCAP-G and hCAP-G-H subcomplexes: wild type (WT), CAP-G K60D/R848E double mutant (K60D/R848E), and CAP-G R168E mutant (R168E). Purified protein samples were subjected to SDS–PAGE and analyzed by CBB staining.

B   Double-stranded DNA (dsDNA) and single-stranded DNA (ssDNA)-binding assay of the hCAP-G. 30-bp dsDNA was incubated with no protein (lanes 1) or increasing amounts of WT hCAP-G (WT; lanes 2–4). 30-mer ssDNA was incubated with no protein (lanes 5) or increasing amounts of WT hCAP-G (WT; lanes 6–8).

C   The dsDNA binding assay for the hCAP-G-H subcomplexes. 30-bp dsDNA was incubated with no protein (lanes 1, 5, and 9), increasing amounts of WT hCAP-G-H subcomplex (WT; lanes 2–4), CAP-G K60D/R848E double mutant (K60D/R848E; lanes 6–8), or CAP-G R168E mutant (R168E; lanes 10–12).

D   The ssDNA binding assay for the hCAP-G-H subcomplexes used in panel (C).

E   The molecular surface of hCAP-G in complex with hCAP-H. The structural model of hCAP-G is shown in white. hCAP-H is shown as a green ribbon model. Identical and homologous residues between hCAP-G and YCG1 are shown in blue and cyan, respectively.

F   Zoomed-in view of the HEPES-binding site. R168 of hCAP-H interacts with HEPES.

(Fig 4C, ΔG). We further confirmed that a ΔG complex harboring the IV-5Q mutations also recapitulated essentially the same phenotype (Fig 4C, ΔG[IV-5Q]). These results strongly suggest that the IV-5Q mutations disrupt both physical and functional interactions between hCAP-G and hCAP-H, resulting in the formation of a tetrameric mutant complex that is equivalent to the ΔG complex.

### Identification of a DNA-binding site in the hCAP-G-H subcomplex

Kschonsak *et al* [17] reported the structure of a YCG1-BRN1-dsDNA ternary complex. Although we carried out numerous trials, we were unable to obtain any crystals of the corresponding ternary complex using hCAP-G and hCAP-H. To examine whether our hCAP-G and hCAP-G-H subcomplex has the ability to interact with DNA, we performed the electrophoretic mobility shift assay (EMSA) using a blunt-ended dsDNA probe and ssDNA probe. hCAP-G did not interact with either dsDNA or ssDNA (Fig 5A and B, lanes 1–8), whereas the hCAP-G-H subcomplex interacted not only with dsDNA (Fig 5A and C, lanes 1–4), but also with ssDNA (Fig 5D, lanes 1–4). These data suggest that hCAP-G-H interaction is required for DNA binding. Next, to clarify the important residues interacting with DNA, we mapped potential DNA-binding residues on the hCAP-G-H subcomplex using structural information from the YCG1-BRN1-dsDNA ternary complex (Fig 5E). The DNA-binding interface of the hCAP-G-H subcomplex was estimated to be similar to that of its budding yeast counterpart. We picked up two positively charged residues (K60 and R848), which correspond to the DNA-binding residues of YCG1 (YC1/2), and constructed a K60D/R848E double mutant. The K60D/R848E double mutation greatly impaired the binding affinity to both dsDNA and ssDNA (Fig 5A, C and D, lanes 5–8). This suggested that dsDNA- and ssDNA-binding interfaces are overlapped, and that the N- and C-terminal HEAT-repeat domains of hCAP-G are likely required for binding to both DNA substrates. The competition assay between ssDNA and dsDNA for the subcomplex binding also supported the above notion (Fig EV4A and B). Next, we mutated R168 of hCAP-G, a residue that is important for HEPES binding (Fig 5F) and potentially confers DNA binding. We found that the R168E mutant reduced, but not completely eliminated, the affinity for both dsDNA and ssDNA (Fig 5A, C and D, lanes 9–12). As the small concave surface containing R168 had insufficient space to accommodate dsDNA, we speculate that it adapts an open-mouth structure to grab dsDNA. Alternatively, a conformational change of this surface may indirectly affect the DNA-binding domain constituted by K60 and R848.

In this study, we determined the crystal structure of an hCAP-G-H subcomplex, in which the kleisin subunit hCAP-H is more loosely bound to the HEAT subunit hCAP-G than the yeast YCG1-BRN1 complex. We also demonstrated the structural basis of the interaction between hCAP-G and hCAP-H, whereby hCAP-H binds to hCAP-G with two conserved N- and C-terminal concave surfaces. Although they have great sequence divergences, the human and yeast structures are similar to each other, suggesting that the basic mechanisms of condensin-mediated chromosome condensation are widely conserved among eukaryotic species with large and small chromosomes. Our functional assay employing *Xenopus* egg extracts demonstrated that the hCAP-G-H interaction is indeed essential for proper mitotic chromosome assembly. It should be noted that the three different mutant complexes lacking hCAP-G (IV-5Q, ΔG and ΔG[IV-5Q]) still retained the ability to be loaded onto chromosomes in our cell-free assay, a result contrary to the prediction from a previous study [17]. Thus, the proposed safety-belt mechanism would not be the sole mechanism that initiates condensin's loading onto DNA or chromatin. It should also be mentioned that, in the previous structural study of the YCG1-BRN1 subcomplex, only one of two molecules in the asymmetric unit formed the BRN1 safety belt, implicating the occurrence of a more flexible and complex set of conformations created by kleisin-HEAT interactions. Although we have been unable to obtain any crystals with a longer stretch hCAP-H, our DNA binding assay clearly showed that hCAP-H is indeed required to make DNA-binding surface together with hCAP-G. It remains unknown whether the ability of the hCAP-G-H subcomplex to interact with ssDNA, which was not observed for its yeast counterpart, may be related to the previously proposed function of condensin I in transcribing regions to regenerate dsDNA [27]. In the future, it will be important to further clarify the similarities and detailed differences in the structure and function of this fundamental chromosome organizing machinery among different eukaryotic species.

## Materials and Methods

### Protein production and purification

cDNAs corresponding to hCAP-G (amino acid residues 1–900) and hCAP-H (amino acid residues 460–515) were cloned into *Bam*HI-*Hind*III and *Nde*I-*Xho*I sites, respectively, of a pETDuet-1 vector (Novagen). Based on the result of secondary structural prediction, we deleted a putative disordered region of CAP-G (residues 479–553) by PCR-based mutagenesis. The final construct, which encoded an N-terminally His$_6$-tagged hCAP-G (residues 1–478, 554–900) and a part of hCAP-H (460–515), was used to transform *Escherichia coli* BL21 (DE3). Cells were grown at 37°C to a cell density of approximately 0.8 at 660 nm in LB medium and then cultured for another ~20 h at 25°C after the addition of 0.2 mM isopropyl β-D-1-thiogalactopyranoside (IPTG). The cells were harvested, resuspended in 10 ml of buffer I (50 mM HEPES-NaOH pH 6.8 and 250 mM NaCl) per gram of cells, and lysed by sonication. The cell lysate was clarified by centrifugation for 1 h at 4°C (48,300× *g*). The supernatant was applied to a 5-ml HiTrap Heparin HP column (GE Healthcare), and the bound proteins were eluted with a linear gradient of 250–800 mM NaCl over a total volume of 95 ml. The collected proteins were diluted with buffer II (50 mM Tris–Hcl pH 8.5) and applied to a 5-ml HiTrap Q HP anion-exchange column (GE Healthcare). The bound proteins were eluted with a linear gradient of 0–600 mM NaCl over a total volume of 95 ml. The eluted proteins were passed through a HiLoad 16/600 Superdex 200 size-exclusion column (GE Healthcare) equilibrated with buffer III (20 mM HEPES-NaOH pH 7.4, 100 mM NaCl and 5 mM DTT), and then concentrated to 15 mg/ml using a Vivaspin (30 kDa MWCO) concentrator (Sartorius). The purity of the hCAP-G-H subcomplex was confirmed by SDS–PAGE followed by Coomassie Brilliant Blue (CBB) staining. The purified protein was frozen with liquid N$_2$ and stored at −80°C until use.

## Crystallization, data collection, and structure determination

Crystallization of the hCAP-G-H subcomplex was performed by the sitting-drop vapor diffusion method using a commercial kit from Hampton Research, Qiagen, and Molecular Dimensions to screen crystallization conditions. Drops were prepared by mixing 0.5 μl of protein solution with 0.5 μl of reservoir solution. Crystals were obtained in a few conditions with polyethylene glycol as a precipitant after a week at 20°C. Conditions were further optimized with the hanging-drop vapor diffusion method. hCAP-G-H subcomplex crystals suitable for X-ray diffraction experiments appeared within 1 week with a reservoir solution consisting of 6.5% (w/v) PEG3350, 0.10 M $MgCl_2$, 0.10 M HEPES-NaOH pH 7.5, and 3% (v/v) ethylene glycol. Heavy atom derivatives of crystals were prepared by the soaking method using a solution of 1 mM potassium dicyanoaurate (I), 7–12% (w/v) PEG3350, 0.10 M $MgCl_2$, and 0.10 M HEPES-NaOH pH 7.5 for 10 min. All crystals were cryoprotected with a reservoir solution including 20–25% (v/v) ethylene glycol before being flash-frozen.

Each crystal was picked up in a nylon loop, and cooled and stored in liquid $N_2$ gas via a Universal V1-Puck (Crystal Positioning System Inc.) until use. X-ray diffraction data of frozen crystals were collected under a stream of $N_2$ gas at −173°C on the BL-17A beamline at Photon Factory (Tsukuba, Japan) using a pixel array photon-counting detector, PILATUS3 S6M (DECTRIS). The hCAP-G-H subcomplex crystal diffracted to 3.0 Å. Diffraction data were integrated, scaled, and averaged with the programs *XDS* [28] and *SCALA* [29].

Initial phases for the Au-labeled hCAP-G-H subcomplex were obtained by single-wavelength anomalous dispersion (SAD) with AutoSol in the *PHENIX* package [30]. Model building of the hCAP-G N-terminal and C-terminal HEAT repeats and hCAP-H was carried out with AutoBuild in *PHENIX*. Subsequent model building, especially HEAT repeats of the middle region of hCAP-G, was performed with COOT [31], and the structure was refined with *PHENIX.REFINE*. The data collection and refinement statistics are summarized in Table 1. All structure drawings in this study were created with PyMOL (http://www.pymol.org/), which depicted a-molecule and b-molecule as representative structures.

## Interaction analysis of hCAP-G and hCAP-H co-expressed in E. coli

cDNA encoding a central part of hCAP-H (residues 394–515) was cloned into the *Nde*I-*Xho*I site of pETDuet-1 containing the cDNA of hCAP-G (residues 1–478, 554–900) in the *Bam*HI-*Hind*III site. Point mutations in the hCAP-G or hCAP-H sequence were introduced using a PCR-based method. $His_6$-tagged hCAP-G was co-expressed with the hCAP-H by a procedure similar to that described above, except that bacterial cells were incubated at 15 or 25°C after IPTG induction. Interaction analysis, based on immobilized metal affinity chromatography (IMAC), was performed. In brief, cell lysates were applied to Ni-NTA agarose resin (Qiagen), and the beads were washed first with buffer IV (50 mM HEPES-NaOH pH 7.4, 1.5 M NaCl, and 20 mM imidazole) and then with buffer V (50 mM HEPES-NaOH pH 7.4 and 100 mM NaCl). The bound proteins were subjected to SDS–PAGE followed by CBB staining. The bands were detected with a ChemiDoc Touch Imaging System (Bio-Rad Laboratories).

## DNA binding assay

To obtain proteins for EMSA, mutations were introduced by the same method described above. hCAP-G alone and hCAP-G-H subcomplex mutants were overexpressed and purified with Ni-NTA agarose resin. The bound protein was washed with buffer IV and buffer V, and then eluted with a stepwise gradient of 50–500 mM imidazole. The eluted proteins were further purified by HiTrap Q and HiLoad Superdex 200. Purified mutant proteins were concentrated, frozen with liquid $N_2$, and stored at −80°C until use.

To investigate the preference of the hCAP-G-H subcomplex for DNA structures, EMSA was performed using 30-mer ssDNA (5′-CCTATAGTGAGTCGTATTACAATTCACTCG-3′) and 30-mer blunt-ended dsDNAs (5′-CCTATAGTGAGTCGTATTACAATTCACTCG-3′; 5′-CGAGTGAATTGTAATACGACTCACTATAGG-3′). The DNA and the subcomplex were mixed at a molar ratio of 1:1, 1:2, or 1:4 and incubated overnight at 4°C. The final concentration of DNA after mixing the solutions was 6.7 μM. These solutions were separated by electrophoresis at 4°C on 1% agarose gel containing GelRed DNA stain (Biotium), and bands were detected by a ChemiDoc Touch Imaging System.

To investigate the competition between dsDNA and ssDNA for hCAP-G-H subcomplex binding, EMSA was performed using 30-mer FAM-ssDNA (FAM-5′-CCTATAGTGAGTCGTATTACAATTCACTCG-3′) and 30-mer blunt-ended dsDNA (5′-CCTATAGTGAGTCGTATTACAATTCACTCG-3′; 5′-CGAGTGAATTGTAATACGACTCACTATAGG-3′). The FAM-ssDNA, dsDNA, and the subcomplex were mixed at a molar ratio of 1:0:0, 1:0:2, 1:1:2, 1:2:2, 1:4:2, 1:6:2, 1:8:2, or 1:10:2 and incubated overnight at 4°C. The final concentration of FAM-ssDNA after mixing the solutions was 1.7 μM. These solutions were separated by electrophoresis at 4°C on 1% agarose gel, and bands were detected with a ChemiDoc Touch Imaging System. To investigate the competition between 30-mer blunt-ended FAM-dsDNA (FAM-5′-CCTATAGTGAGTCGTATTA-CAATTCACTCG-3′; 5′-CGAGTGAATTGTAATACGACTCACTATAGG-3′) and 30-mer ssDNA (5′-CCTATAGTGAGTCGTATTACA ATTCACTCG-3′) for the subcomplex binding, we used the same method.

## Expression and purification of recombinant condensin complexes

To construct the IV-5Q mutant of hCAP-H, we used the Quik-Change Site-Directed Mutagenesis Kit (Agilent Technologies) to introduce a set of point mutations sequentially into the original expression construct (pFH101) [4] such that five amino acids (F463, F469, F473, F501, and Y503) in its coding sequence were substituted with glutamine (Q). The oligonucleotides used for mutagenesis were as follows (mutation sites introduced are underlined): F469Q, 5′-GAAGATGATATTGAC<u>CAA</u>GATGTATATT TTAGA-3′; F501Q Y503Q, 5′-CCTTCCTACAGAT<u>CAAAAC</u><u>CAGA</u> ATGTTGACACTCT-3′; F463Q, 5′-GATTTTGAAATTGAC<u>CAA</u>GA AGATGATATTGAC-3′; F469Q F473Q, 5′-GAC<u>CAA</u>GATGTATA TC<u>AAA</u>GAAAAACAAAGGCT-3′. The resultant construct (pHM110) was used for the preparation of bacmid DNA to produce a baculovirus. Expression of condensin holocomplexes and subcomplexes in insect cells, and their purification were performed as described previously [18].

**Chromosome assembly assays and immunofluorescence analyses**

Chromosome assembly assays using *Xenopus* egg extracts and immunofluorescence analyses of chromosomes assembled in the extracts were performed as described previously [18].

## Data availability

The coordinates for the structures reported in this paper have been deposited in PDB under the accession number 6IGX.

**Expanded View** for this article is available online.

### Acknowledgements

We acknowledge the kind support of the beamline staff of Photon Factory for data collection. We also thank Dr. T. Nagashima, Ms. K. Yoda, and Mr. K. Ishiguchi for their technical assistance. This work was supported by the Japan Society for the Promotion of Science (JSPS), KAKENHI (grant nos. 15K18491 and 17K07314 to KH, 15K06959 to KK, 16H04755 and 17H06014 to HH, and 15H05971 to TH), and grants from the Takeda Science Foundation (HH) and the Naito Foundation (HH).

### Author contributions

KH and TH designed the experiments. KH, TM, KM, and KS produced the recombinant proteins. KH, TM, and KS carried out the crystallization and structural determination. KH and KM carried out the *in vitro* interaction assay. KH and KM carried out the DNA binding assay. KT performed mutagenesis of hCAP-H, and KK purified the recombinant holocomplexes and performed functional assays using *Xenopus* egg cell-free extracts. KH, TH, and HH wrote the manuscript.

### Conflict of interest

The authors declare that they have no conflict of interest.

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
