## [Review Process File · EMBO Reports]

Structural basis of HEAT-kleisin interactions in the human condensin I subcomplex

Kodai Hara, Kazuhisa Kinoshita, Tomoko Migita, Kei Murakami, Kenichiro Shimizu, Kozo Takeuchi, Tatsuya Hirano, and Hiroshi Hashimoto

Review timeline:	Submission date:	3 October 2018
	Editorial Decision:	26 October 2018
	Revision received:	20 January 2019
	Editorial Decision:	6 February 2019
	Revision received:	13 February 2019
	Accepted:	20 February 2019

Editor: Achim Breiling

Transaction Report:

1st Editorial Decision

26 October 2018

Thank you for the submission of your research manuscript to EMBO reports. We have now received reports from the three referees that were asked to evaluate your study, which can be found at the end of this email.

As you will see, all referees think the manuscript is of interest, but requires a major revision to allow publication in EMBO reports. All three referees have a number of concerns and/or suggestions to improve the manuscript, which we ask you to address in a revised manuscript. As the reports are below, I will not detail them here. Further, as indicated by all the referees, the paper needs also revision regarding grammar and language (to also render it more comprehensible for readers not from the immediate field - see also point 6 by referee #2), and finally needs to be proofread by a native speaker.

Given the constructive referee comments, I would like to invite you to revise your manuscript with the understanding that all referee concerns must be addressed in the revised manuscript and/or in a detailed point-by-point response. Acceptance of your manuscript will depend on a positive outcome of a second round of review. It is EMBO reports policy to allow a single round of revision only and acceptance or rejection of the manuscript will therefore depend on the completeness of your responses included in the next, final version of the manuscript.

Revised manuscripts should be submitted within three months of a request for revision; they will otherwise be treated as new submissions. Please contact us if a 3-months time frame is not sufficient for the revisions so that we can discuss the revisions further.

Supplementary/additional data: The Expanded View format, which will be displayed in the main HTML of the paper in a collapsible format, has replaced the Supplementary information. You can submit up to 5 images as Expanded View. Please follow the nomenclature Figure EV1, Figure EV2 etc. The figure legend for these should be included in the main manuscript document file in a section called Expanded View Figure Legends after the main Figure Legends section. Additional Supplementary material should be supplied as a single pdf labeled Appendix. The Appendix includes a table of content on the first page, all figures and their legends. Please follow the nomenclature Appendix Figure Sx throughout the text and also

label the figures according to this nomenclature.

For more details please refer to our guide to authors:

<http://embor.embopress.org/authorguide#manuscriptpreparation>

Important: All materials and methods should be included in the main manuscript file.

See also our guide for figure preparation:

http://www.embopress.org/sites/default/files/EMBOPress_Figure_Guidelines_061115.pdf

Regarding data quantification and statistics, can you please specify, where applicable, the number "n" for how many independent experiments (biological replicates) were performed, the bars and error bars (e.g. SEM, SD) and the test used to calculate p-values in the respective figure legends. Please provide statistical testing where applicable. See:

<http://embor.embopress.org/authorguide#statisticalanalysis>

Please also format the references according to EMBO reports style. See:

<http://embor.embopress.org/authorguide#referencesformat>

We now strongly encourage the publication of original source data with the aim of making primary data more accessible and transparent to the reader. The source data will be published in a separate source data file online along with the accepted manuscript and will be linked to the relevant figure. If you would like to use this opportunity, please submit the source data (for example scans of entire gels or blots, data points of graphs in an excel sheet, additional images, etc.) of your key experiments together with the revised manuscript. Please include size markers for scans of entire gels, label the scans with figure and panel number, and send one PDF file per figure.

- a complete author checklist, which you can download from our author guidelines (<http://embor.embopress.org/authorguide#revision>). Please insert page numbers in the checklist to indicate where the requested information can be found.
- a letter detailing your responses to the referee comments in Word format (.doc)
- a Microsoft Word file (.doc) of the revised manuscript text
- editable TIFF or EPS-formatted single figure files in high resolution (for main figures and EV figures)

I look forward to seeing a revised version of your manuscript when it is ready. Please let me know if you have questions or comments regarding the revision.

REFeree REPORTS

Referee #1:

The manuscript by Hara et al. describes a crystal structure of a sub-complex of human condensin as well as follow-up mutational analysis using reconstituted protein complexes. The structure of hCAP-G bound to a fragment of hCAP-H reveals a harp-like shape with hCAP-H sequences found at the central cleft - as recently seen in related budding and fission yeast crystal structures. The authors describe in quite some detail residues located at the extended interface between hCAP-G and hCAP-H and generate mutants that are defective in the CAP-G/CAP-H interaction and in DNA binding. The mutants are further characterized by gel-shift experiments and chromosome condensation assays in the *Xenopus* egg extract system.

The novelty of the presented study is somewhat compromised by the availability of related structures including DNA-protein co-crystal structures. Nevertheless, the study offers important insights. The new structure of a human condensin sub-complex will be an invaluable tool for the design of mutational studies and for comparative analysis. The DNA binding studies show that CAP-G/CAP-H fails to discriminate between single- and double-stranded DNA, which may be relevant for condensin recruitment to transcribed regions. And most importantly, the reconstitution experiments indicate that the CAP-G/CAP-H association might be dispensable for chromosome association of condensin and for the formation of (fuzzy) mitotic chromosomes (presumably via DNA loop extrusion). The last point is very important in the light of other

recent studies. It should be supported by additional experiments (see below) and more prominently discussed and stressed in the manuscript.

All data appear very solid, and the text is presented well. The latter may benefit from shortening (see below). Altogether, the manuscript will be a valuable addition to the SMC literature, once the following points have been considered.

Major point

In many recent models, DNA loop extrusion initiates with the binding of condensin to chromosomal DNA via the CAP-G/H DNA-safety-belt. The work presented in this manuscript (1) and the data in Kinoshita et al., 2015 (2) challenge this notion and highlight the potential dispensability of the DNA-safety-belt for partial chromosome condensation. However, the two studies have potential weaknesses: (1) The hCAP-H(IV-5Q) mutant may co-purify residual hCAP-H and (2) residual *Xenopus* CAP-G (due to incomplete depletion) may partially compensate for the absence of hCAP-G. To eliminate these possibilities, it is highly desirable to combine the two mutations and reconstitute condensin with CAP-H IV-5Q subunits and lacking hCAP-G subunits. If condensin is able to support the dynamic assembly of chromosomal axes in the complete absence of CAP-G, then the double mutant complex should produce (fuzzy) chromatids similar to the two single mutants (Figure 4C).

Minor points

Please give details (including background subtraction and standard curves) on the quantification of CBB stained bands (Fig. 3B). It seems difficult to reliably quantify high-intensity CAP-G and barely detectable CAP-H bands from the same gel.

The description of the composite interface between CAP-G and CAP-H in the main text is very detailed and may benefit from shortening and focussing. Parts of the description may be moved to supplemental material. Also, occasionally, functional implications are drawn from residue arrangement without additional experimental evidence. For example, "the conserved L508 of hCAP-H contributes to anchor the $\alpha 4$ helix to a hydrophobic cleft." Please rewrite this and similar statements.

Figure 1A: Could the scheme indicate the exact constructions used for crystallography and the residues visible in the structure?

When discussing structural similarity between human and yeast G/H complexes it would be useful to mention the RMSD for the two yeast structures for comparison. Also, the similarity and differences with the DNA bound and DNA free structures should be discussed explicitly.

Figure panel 3B is called before 3A in the main text.

----- Referee #2:

This manuscript describes the crystal structure and accompanying biochemical and cell biological experiments of the human condensin HEAT repeat subunit CAP-G bound to a portion of the kleisin subunit CAP-H. The structure is nicely done and the accompanying experiments are well performed and support the authors conclusions. There are obvious similarities to the structure of the yeast orthologues Ycg1-Brn1, published last year. There are also several distinct differences. Some of the differences are small, but others are striking, for example the ability of condensin to bind to DNA without the need to recruit CAP-G and the ability to interact with single stranded DNA. The manuscript therefore reports the first structure of a human condensin subunit and it expands our knowledge on a few important aspects of condensin function. The overall conceptual advance is moderate, but I can in principle recommend this manuscript for publication. The authors should consider the following points during the preparation of a revised manuscript.

In the order of appearance rather than importance:

1. page 3: Reference Wilhelm et al. 2015 is missing from the reference list.
2. page 5: 'We found that the entire HEAT domain of hCAP-G lacking the internal disordered region...' It

remains unclear how this was achieved. Were two separate hCAP-G fragments expressed? Were they connected by a smaller linker? This should be clarified.

3. page 5: The authors structure shows that in addition to the predicted disordered H12 loop in hCAP-G, there is an additional, smaller disordered loop in repeat H15. Looking back at their PrDOS sequence analysis, was there any sign of this H15 loop?

4. page 6: '...four basic residues [...] are also conserved'. This is shown on the sequence alignment in Fig EV1A. It would be nice to present a small overlay of a yeast and the human structure that shows this conservation.

5. page 7: '...corresponds to the buckle region...', it is not clear to the non-expert reader what a buckle region is. This should be explained, taking into account that the yeast seatbelt mechanism is rather controversial. See also below.

6. A lot of the manuscript is written in crystallography slang that is inaccessible to a molecular biologist or cell biologist. Please rewrite these sections with help of one of the cell biology co-authors to make them more accessible. E.g. page 7 'distance between the NH2 of R257 and the CG2 of V754 of hCAP-G, located at H7 and H16, respectively, is 17.06 Å'. This sentence contains important information as to the width of the harp, but remains cryptic to a general reader.

7. Related to the point above, the authors suggest that their structure presents an 'open' conformation of hCAP-G before DNA binding. However, in case of the yeast structure, there is little conformational difference between the structures without or with DNA. DNA binding is therefore not a likely reason for the difference in the width of the harp. Rather, these SMC HEAT repeat subunits appear to be generally very flexible. This has probably been best documented in case of the cohesin HEAT subunit Scc2 by electron microscopy (Chao et al. 2015). Any crystal is likely, depending on the conditions used, to capture one of probably many possible conformations. This should be discussed.

8. One difference from the yeast Ycg1 structure is the Brn1 'seatbelt' that has been proposed in the latter. This is the most controversial aspect of the yeast structure as it is only seen in one of two molecules in the asymmetric unit and because one of the conserved hydrophobic residues engages in a crystal contact to form the 'seatbelt'. The seatbelt is therefore likely a crystal artefact. It would be important to discuss this or hear of any information that emerged from the authors own studies that addresses the possible conformation of farther N-terminal kleisin sequences. Have the authors tried to co-crystallize a longer stretch of CAP-H?

9. With respect to the above, the authors' results that positively charged residues on various surfaces of CAP-G contribute to DNA binding is interesting and relevant. It suggests that DNA binding by CAP-G is more varied than the seatbelt model suggests. Rather, it seems that the HEAT subunits provide a series of DNA contacts that probably direct DNA towards its final resting place that, as the authors point out, lies within the condensin ring. This is worth to discuss.

10. Figure 1B, can CAP-H helix alpha1 be shown in this structure, maybe in the 90 degree rotated view?

11. Figure 1B illustrates the positions of disordered loops in both CAP-H and CAP-G. It would be very informative to draw these loops approximately to scale, estimated by the length of a disordered polypeptide of the given length. It is important to get a sense of how much peptide sequence lies within each loop.

12. Figure 1D. A stereoview is not helpful for the general reader. Instead a larger view of this superimposition, with depth added by graphic means, would be more informative.

Referee #3:

Hashimoto and colleagues report on the crystal structure of human CAP-G in complex with a fragment of CAP-H. They found that the overall structure is similar to the corresponding yeast complex, with a few notable differences. Pull-down assays of native and mutants were used to determine protein-protein interactions. Mutant CAP-G is not assembled into the holocomplex and these complexes when addback to

Xenopus extracts bind to chromatin but fail to condense it. Finally, the authors demonstrate that the complex bind both single and double stranded DNA. The crystallography data is solid. However, conclusions on flexibility are not fully supported by the data. Additional pull-down and gel shift experiments are required in order to clarify the biological activity. The text requires extensive language editing as many typos and unclear sentences are found throughout the text.

Major concerns:

1. The auteurs speculate that the CAP-G-H is flexible and change conformation upon DNA binding. The b-factors should be shown and discussed. This possibility can be tested experimentally with the available reagents by CD or limited proteolysis.
2. The authors claim that the F501 and Y503 are required for stabilization. there is no significant difference between the 3 and 5 series mutants. Mutants that carry substitutions of these 2 specific residues need to be tested and compared to the 3 mutants.
3. In order to determine if single and double stranded DNA share the same binding site competition experiments should be done. Increased concentration of dsDNA should be added to the ssDNA binding reaction and vice versa.
4. The effect of CAP-G-H interaction on DNA binding should be tested (e.g. CAP-G D647K)

Minor points:

1. English editing throughout the text
2. A control showing uninduced cells is needed for evaluation of the bands in figures 3C-D
3. The CAP-G-H complexes were incubated with DNA overnight. Is binding kinetics very slow?

1st Revision - authors' response

20 January 2019

Referee #1

1) In many recent models, DNA loop extrusion initiates with the binding of condensin to chromosomal DNA via the CAP-G/H DNA-safety-belt. The work presented in this manuscript (1) and the data in Kinoshita et al., 2015 (2) challenge this notion and highlight the potential dispensability of the DNA-safety-belt for partial chromosome condensation. However, the two studies have potential weaknesses: (1) The hCAP-H(IV-5Q) mutant may co-purify residual hCAP-H and (2) residual Xenopus CAP-G (due to incomplete depletion) may partially compensate for the absence of hCAP-G. To eliminate these possibilities, it is highly desirable to combine the two mutations and reconstitute condensin with CAP-H IV-5Q subunits and lacking hCAP-G subunits. If condensin is able to support the dynamic assembly of chromosomal axes in the complete absence of CAP-G, then the double mutant complex should produce (fuzzy) chromatids similar to the two single mutants (Figure 4C).

Response:

We thank the reviewer for bringing up this important issue. According to the reviewer's comment, we purified a DG tetramer complex harboring the IV-5Q mutations and tested for its ability to assemble chromosomes in the cell-free extract. The "double" mutant complex also produced fuzzy chromatids similar to the two "single" mutant complexes, further substantiating our conclusion that the CAP-G subunit is not necessarily essential for condensin I to be loaded onto chromosomes. Our results strongly suggest that the proposed safety belt mechanism is not the sole mechanism that initiates condensin's loading onto DNA or chromatin, and that there must be another pathway for its loading. We have added the results to Fig. 4C and revised the text on page 11.

2) Please give details (including background subtraction and standard curves) on the quantification of CBB stained bands (Fig. 3B). It seems difficult to reliably quantify high-intensity CAP-G and barely detectable CAP-H bands from the same gel.

Response:

We measured the band intensities of hCAP-G and its mutant forms, and subtracted a background intensity from them. We also obtained the band intensities of hCAP-H in the same manner. The relative band intensities were obtained by dividing the intensities of hCAP-H by those of hCAP-G, and normalized by the values of wild type. We did not use the standard curve for the calculation. That said, we are aware that this type of interaction analysis based on IMAC-CBB staining is convenient but is not very quantitative. We have therefore decided to remove the graph in Fig. 3C from the revised manuscript.

3) The description of the composite interface between CAP-G and CAP-H in the main text is very detailed and may benefit from shortening and focussing. Parts of the description may be moved to supplemental material. Also, occasionally, functional implications are drawn from residue arrangement without additional experimental evidence. For example, "the conserved L508 of hCAP-H contributes to anchor the α 4 helix to a hydrophobic cleft." Please rewrite this and similar statements.

Response:

We have revised the text on pages 7-9. The current manuscript especially focuses on the amino acid residues of hCAP-H responsible for hCAP-G binding.

4) Figure 1A: Could the scheme indicate the exact constructions used for crystallography and the residues visible in the structure?

Response:

We have added a scheme that indicates the exact constructions used for crystallography in Fig. 1B and renumbered Fig. 1. We have also added the residue numbers for H12 and H15 loops in Fig. 1C.

5) When discussing structural similarity between human and yeast G/H complexes it would be useful to mention the RMSD for the two yeast structures for comparison. Also, the similarity and differences with the DNA bound and DNA free structures should be discussed explicitly.

Response:

We have mentioned the RMSD for the two yeast structures on pages 6-7 in the revised manuscript. We have also added the superimpositions between DNA bound and DNA free structures in Fig. 1E and discussed the similarity and differences between the DNA-bound and DNA-free structures on page 7.

6) Figure panel 3B is called before 3A in the main text.

Response:

We would like to thank the reviewer for pointing out this. In the revised manuscript, we have modified Figure 3 so that panel A is called before panel B in the main text.

Referee #2

1) page 3: Reference Wilhelm et al. 2015 is missing from the reference list.

Response:

We would like to thank the reviewer for pointing out our error. We have added this missing reference to the reference list.

2) page 5: 'We found that the entire HEAT domain of hCAP-G lacking the internal disordered region...' It remains unclear how this was achieved. Were two separate hCAP-G fragments expressed? Were they connected by a smaller linker? This should be clarified.

Response:

The N-terminal fragment of hCAP-G (residues 1-478) connected with its C-terminal fragment (residues 554-900) was co-expressed and co-purified with a fragment of hCAP-H containing motif IV, and used for crystallography. We have revised the text accordingly on page 5.

3) page 5: The authors structure shows that in addition to the predicted disordered H12 loop in hCAP-G, there is an additional, smaller disordered loop in repeat H15. Looking back at their PrDOS sequence analysis, was there any sign of this H15 loop?

Response:

We examined the PrDOS sequence analysis for hCAP-G. The analysis predicted 7 disordered loops including residues 1-12, 81-93, 382-393, 477-553 (H12 loop), 660-687 (H15 loop), 812-821, & 896-1015. Thus, the analysis also predicted the H15 loop. We have revised the text accordingly on page 5.

4) page 6: '...four basic residues [...] are also conserved'. This is shown on the sequence alignment in Fig EV1A. It would be nice to present a small overlay of a yeast and the human structure that shows this conservation.

Response:

Our crystallized structure did not include these residues. We apologize that our original explanation created an unnecessary confusion. We have revised the text on page 6.

5) page 7: '...corresponds to the buckle region...', it is not clear to the non-expert reader what a buckle region is. This should be explained, taking into account that the yeast seatbelt mechanism is rather controversial. See also below.

Response:

We have included an additional explanation about the yeast seatbelt mechanism on page 6.

6) A lot of the manuscript is written in crystallography slang that is inaccessible to a molecular biologist or cell biologist. Please rewrite these sections with help of one of the cell biology co-authors to make them more accessible. E.g. page 7 'distance between the NH2 of R257 and the CG2 of V754 of hCAP-G, located at H7 and H16, respectively, is 17.06 Å'. This sentence contains important information as to the width of the harp, but remains cryptic to a general reader.

Response:

We have revised the text on pages 5-9.

7) Related to the point above, the authors suggest that their structure presents an 'open' conformation of hCAP-G before DNA binding. However, in case of the yeast structure, there is little conformational difference between the structures without or with DNA. DNA binding is therefore not a likely reason for the difference in the width of the harp. Rather, these SMC HEAT repeat subunits appear to be generally very flexible. This has probably been best documented in case of the cohesin HEAT subunit Scc2 by electron microscopy (Chao et al. 2015). Any crystal is likely, depending on the conditions used, to capture one of probably many possible conformations. This should be discussed.

Response:

We have revised the text on page 7 and added the report by Chao et al. to the reference list to describe the flexibility of HEAT subunits.

8) One difference from the yeast Ycg1 structure is the Brn1 'seatbelt' that has been proposed in the latter. This is the most controversial aspect of the yeast structure as it is only seen in one of two molecules in the asymmetric unit and because one of the conserved hydrophobic residues engages in a crystal contact to form the 'seatbelt'. The seatbelt is therefore likely a crystal artifact. It would be important to discuss this or hear of any information that emerged from the authors own studies that addresses the possible conformation of farther N-terminal kleisin sequences. Have the authors tried to co-crystallize a longer stretch of CAP-H?

Response:

We would like to thank the reviewer for pointing out this important issue. In the revised manuscript, we have added the above discussion on pages 12-13. We attempted to co-crystallize a longer stretch of hCAP-H with hCAP-G, but no crystals have been obtained. We will try to clarify the role of further N-terminal kleisin sequences in future studies.

9) With respect to the above, the authors' results that positively charged residues on various surfaces of CAP-G contribute to DNA binding is interesting and relevant. It suggests that DNA binding by CAP-G is more varied than the seatbelt model suggests. Rather, it seems that the HEAT subunits provide a series of DNA contacts that probably direct DNA towards its final resting place that, as the authors point out, lies within the condensin ring. This is worth to discuss.

Response:

We have revised the text on page 13.

10) Figure 1B, can CAP-H helix $\alpha 1$ be shown in this structure, maybe in the 90 degree rotated view?

Response:

We were unable to show the $\alpha 1$ helix of hCAP-H in Figure 1B, because our structure did not include this helix. We tried to crystallize a hCAP-G-H complex that includes the $\alpha 1$ helix of hCAP-H, but we were unable to obtain any crystals.

11) Figure 1B illustrates the positions of disordered loops in both CAP-H and CAP-G. It would be very informative to draw these loops approximately to scale, estimated by the length of a disordered polypeptide of the given length. It is important to get a sense of how much peptide sequence lies within each loop.

Response:

We have adjusted the size of the H15 loop to scale in Fig. 1C (Fig. 1 was renumbered).

12) Figure 1D. A stereoview is not helpful for the general reader. Instead a larger view of this superimposition, with depth added by graphic means, would be more informative.

Response:

We have revised **Fig. 1D**.

Referee #3

1) The authors speculate that the CAP-G-H is flexible and change conformation upon DNA binding. The b-factors should be shown and discussed. This possibility can be tested experimentally with the available reagents by CD or limited proteolysis.

Response:

We have added the b-factors of the hCAP-G-H subcomplex with its related structures in **Fig. EV2C**, and discussed the flexibility of hCAP-G-H using the b-factors on page 7. The b-factors of the middle region of hCAP-G are higher than those of other regions, indicating that hCAP-G-H is flexible and changes its conformation upon DNA binding.

2) The authors claim that the F501 and Y503 are required for stabilization. There is no significant difference between the 3 and 5 series mutants. Mutants that carry substitutions of these 2 specific residues need to be tested and compared to the 3 mutants.

Response:

We prepared F501A/Y503A (2A) double mutants and added the results to **Fig. 3C, lane 11**. The data shows that F501 and Y503 were not required for the complex formation of hCAP-G-H. We have revised the text accordingly on page 10.

3) In order to determine if single and double stranded DNA share the same binding site competition experiments should be done. Increased concentration of dsDNA should be added to the ssDNA binding reaction and vice versa.

Response:

We performed the competition assay between ssDNA and dsDNA, and added its result to **Fig. EV4**. These data support that both DNA binding sites are overlapped. We have revised the text on page 12.

4) The effect of CAP-G-H interaction on DNA binding should be tested (e.g. CAP-G D647K)

Response:

According to the reviewer's suggestion, we tried to purify hCAP-G D647K, but it was difficult to obtain a sufficient amount of recombinant protein. Instead, we purified the wild-type hCAP-G alone (without hCAP-H) and tested for its DNA binding activity (**Fig. 5A-B**). The data shows that hCAP-G alone binds neither dsDNA nor ssDNA, suggesting that the hCAP-G-H interaction is required for DNA binding. We have revised the text accordingly on page 11.

5) English editing throughout the text

Response:

The revised manuscript has been proofread by a native speaker.

6) A control showing uninduced cells is needed for evaluation of the bands in figures 3C-D

Response:

We have added an "uninduced" control to **Fig. 3C, lane 9**.

7) The CAP-G-H complexes were incubated with DNA overnight. Is binding kinetics very slow?

Response:

Although DNA binding by the hCAP-G-H complexes was detectable in a shorter incubation time, we felt that the binding affinity was not high. We therefore performed the binding assay in the long incubation time. The detailed binding kinetics will be tested in future studies.

Thank you for the submission of your revised manuscript to our editorial offices. We have now received the reports from the three referees that were asked to re-evaluate your study, you will find below. As you will see, the referees now support the publication of your manuscript in EMBO reports.

Before we can proceed with formal acceptance, I have these few editorial requests, which we ask you to address in a final revised version of the manuscript:

- Please provide the abstract written in present tense.
- Please find attached a word file of the manuscript text (provided by our publisher) with changes we ask you to include in your final manuscript text, and some queries, we ask you to address. Please provide your final manuscript file with track changes, in order that we can see the modifications done.

Further, I would need from you:

- a short, two-sentence summary of the manuscript
- two to three bullet points highlighting the key findings of your study
- a schematic summary figure (in jpeg or tiff format with the exact width of 550 pixels and a height of not more than 400 pixels) that can be used as a visual synopsis on our website.

REFEREE REPORTS

Referee #1:

The authors have appropriately responded to the comments of the reviewers. The manuscript has been significantly improved. It should be published without further delay.

Referee #2:

I read the revised manuscript. The authors have made further improvements that make this a very important study that combines structural and cell biology to extend our molecular understanding of condensin. It also clarifies important aspects that were left unclear in previous studies. I can highly recommend it for publication.

Referee #3:

In the revised manuscript the authors addressed all my concerns and I support its publication.

2nd Revision - authors' response

13 February 2019

The authors performed all minor editorial changes.

Corresponding Author Name: Kodai Hara & Tatsuya Hirano

Manuscript Number: EMBOR-2018-47183V3